# Definitive Cefazolin Therapy for Stabilized Adults with Community-Onset *Escherichia coli*, *Klebsiella* Species, and *Proteus mirabilis* Bacteremia: MIC Matters

**DOI:** 10.3390/jcm9010157

**Published:** 2020-01-07

**Authors:** Chih-Chia Hsieh, Po-Lin Chen, Chung-Hsun Lee, Chao-Yung Yang, Ching-Chi Lee, Wen-Chien Ko

**Affiliations:** 1Department of Emergency Medicine, National Cheng Kung University Hospital, College of Medicine, National Cheng Kung University, Tainan 70403, Taiwan; hsiehchihchia@gmail.com (C.-C.H.); chlee82er@yahoo.com.tw (C.-H.L.); chao.youg@gmail.com (C.-Y.Y.); 2Department of Internal Medicine, National Cheng Kung University Hospital, College of Medicine, National Cheng Kung University, Tainan 70403, Taiwan; cplin@mail.ncku.edu.tw; 3Department of Adult Critical Care Medicine, Tainan Sin-Lau Hospital, Tainan 70142, Taiwan; 4Graduate Institute of Medical Sciences, College of Health Sciences, Chang Jung Christian University, Tainan 71101, Taiwan; 5Department of Medicine, College of Medicine, National Cheng Kung University, Tainan 70101, Taiwan

**Keywords:** cefazolin, definitive therapy, bacteremia, *Escherichia coli*, *Klebsiella pneumoniae*, *Proteus mirabilis*

## Abstract

Background: Cefazolin is in vitro active against wild isolates of *Escherichia coli*, *Klebsiella* species, and *Proteus mirabilis* (EKP), but clinical evidence supporting the contemporary susceptibility breakpoint issued by the Clinical and Laboratory Standards Institute (CLSI) are limited. Methods: Between 2010 and 2015, adults with monomicrobial community-onset EKP bacteremia with definitive cefazolin treatment (DCT) at two hospitals were analyzed. Cefazolin minimum inhibitory concentrations (MICs) were correlated with clinical outcomes, including primary (treatment failure of DCT) and secondary (30-day mortality after bacteremia onset, recurrent bacteremia, and mortality within 90 days after the end of DCT) outcomes. Results: Overall, 466 bacteremic episodes, including 340 (76.2%) episodes due to *E. coli*, 90 (20.2%) *Klebsiella* species, and 16 (3.6%) *P. mirabilis* isolates, were analyzed. The mean age of these patients was 67.8 years and female-predominated (68.4%). A crude 15- and 30-day mortality rate was 0.7% and 2.2%, respectively, and 11.2% experienced treatment failure of DCT. A significant linear-by-linear association of cefazolin MICs, with the rate of treatment failure, 30-day crude mortality, recurrent bacteremia or 90-day mortality after the DCT was present (all *γ* = 1.00, *p* = 0.01). After adjustment, the significant impact of cefazolin MIC breakpoint on treatment failure and 30-day crude mortality was most evident in 2 mg/L (>2 mg/L vs. ≤2 mg/L; adjusted hazard ratio, 3.69 and 4.79; *p* < 0.001 and 0.02, respectively). Conclusion: For stabilized patients with community-onset EKP bacteremia after appropriate empirical antimicrobial therapy, cefazolin might be recommended as a definitive therapy for cefazolin-susceptible EKP bacteremia, based on the contemporary CLSI breakpoint.

## 1. Introduction

Cefazolin, a parental first-generation cephalosporin, is bactericidal against *Staphylococcus aureus*, streptococci, *Escherichia coli*, *Klebsiella* species, and *Proteus mirabilis* (EKP) [1]. EKP are the common pathogens that cause varied infections in the community, such as urinary tract infections [2], biliary tract infections [3], and bacteremia [3,4]. However, most published clinical studies have focused on its efficacy in surgical prophylaxis [5] and infections predominately due to staphylococci, such as bloodstream infections [6], bone and joint infections [7], skin and skin structure infections [8], and peritonitis related to continuous ambulatory peritoneal dialysis [9].

Bacteremia is associated with high morbidity and mortality that results in considerable health care expenditure [10]. The Clinical and Laboratory Standards Institute (CLSI) revised the cefazolin interpretive criteria for Enterobacteriaceae isolates in 2012, based on in vitro susceptibility, pharmacokinetic–pharmacodynamic analyses, and limited clinical outcome data [11,12]. The susceptibility breakpoint for cefazolin was reduced from a minimum inhibitory concentration (MIC) of ≤8 to ≤2 mg/L. Furthermore, cefazolin interpretive breakpoints for *Enterobacteriaceae* bacteremia are not documented by the European Committee on Antimicrobial Susceptibility Testing (EUCAST) [13]. Although the impact of the revised susceptibility breakpoint on patient outcomes has been discussed in the literature [14,15], comprehensive clinical data supporting the contemporary revision of bloodstream infections are not evident. Therefore, to provide the rationale of the MIC breakpoint revision, we analyzed clinical characteristics and outcomes of adults with EKP bacteremia definitively treated by cefazolin treatment.

## 2. Methods

### 2.1. Study Design and Population

A retrospective cohort study was conducted during a six year period (2010–2015) at emergency departments (EDs) of two hospitals in southern Taiwan. One hospital is a university-affiliated medical center with 1300 beds and another is a teaching hospital with 800 beds. The study was approved by the institutional review board of National Cheng Kung University Hospital (ER-100-182) and Sin-Lau Hospital (SLH 9919-108-006), and reported by the format recommended by the STROBE (Strengthening the Reporting of Observational Studies in Epidemiology) [16]. Partial clinical information in this cohort has been published [17,18].

Adults with blood cultures sampled at EDs between January 2010 and December 2015 were screened for bacterial growth in blood cultures. For adults with monomicrobial EKP bacteremia, medical information was retrieved from medical records using a predetermined form. If there were multiple bacteremic episodes in a patient, only the first episode was included. Only adults with community-onset bacteremia, who initiated with cefazolin therapy within 3–5 days after bacteremia onset or were treated with cefazolin for the entire antimicrobial course, were included. Patients were excluded if they had hospital-onset bacteremia, received inadequate empirical therapy, were directly discharged from the ED (i.e., were not hospitalized), died within three days after bacteremia onset, had not received definitive cefazolin therapy with the appropriate route and dosage (2 g, every 8 h intravenously), or the information for the clinical outcome was incomplete.

### 2.2. Data Collection

Clinical variables, including age, gender, vital signs and laboratory data at ED, comorbidities, comorbidity severity (McCabe classification), duration and type of antimicrobial agents, bacteremia source, bacteremia severity (a Pitt bacteremia score), duration of hospital stay, and patient outcomes, were retrospectively collected by reviewing medical records of all eligible patients. Two of the authors were randomly assigned to review medical records. Based on cefazolin MICs of the causative bacteremic isolates, included patients were stratified into four categories: ≤1 mg/L, 2 mg/L, 4 mg/L, and 8–16 mg/L.

### 2.3. Patient Outcomes

Bacteremia severity at 72 h after bacteremia onset was regarded as the baseline for the initiation of definitive therapy. The primary outcome was assessed as treatment failure of definitive cefazolin therapy at the 3 to 15 day visit after bacteremia onset, and included a composite of antimicrobial escalation to broad-spectrum agents, the development of breakthrough bacteremia, or the need for intensive care during definitive cefazolin therapy and 15-day crude mortality after bacteremia onset. The secondary outcomes, including 30-day crude mortality after bacteremia onset, recurrent infections, and fatal outcomes within 90 days after the end of definitive cefazolin therapy, were assessed during the period between Day 15 of the bacteremia episode and the 90 day visit after the end of definitive cefazolin therapy.

### 2.4. Microbiological Methods

EKP isolates were identified by the Gram-Negative-Identification Card of the Vitek system (bioMe’rieux, Lyon, France). During the study period, these isolated from blood culture were prospectively stored. Cefazolin MICs was manually determined by the broth microdilution method. Susceptibility to empirical antimicrobials was tested by the disk diffusion method. The antimicrobial susceptibility was interpreted based on contemporary CLSI breakpoints [12].

### 2.5. Definitions

Community-onset bacteremia indicates that the place of bacteremia onset is the community, and includes long-term healthcare, facility-acquired and community-acquired bacteremia, as previously described [4,19]. Since susceptibility data were available approximately three days after bacteremia onset, empirical therapy was arbitrarily defined as the drugs prescribed within three days after bacteremia onset, whereas definitive therapy referred to the drugs prescribed when the susceptibility result became available. As previously described [4,19], antimicrobial therapy was considered to be appropriate when the following two criteria were fulfilled: (i) the route and dosage of antimicrobial administration were as recommended in the Sanford Guide [20]; and (ii) causative pathogens exhibited in vitro susceptibility to the administrated drugs according to the contemporary CLSI breakpoint [12]. The time-to-appropriate antibiotic measured in hours was defined as the period between bacteremia onset (i.e., ED arrival) and administration of the first dose of appropriate antimicrobials. A time-to-appropriate antibiotic of >24 h was considered as inappropriate empirical therapy [4,21].

To assess the disease severity at bacteremia onset and Day 3 after bacteremia onset (i.e., the initiation of definitive antibiotic therapy), a Pitt bacteremia score, which is a validated score based on vital signs, vasopressor agent use, mental status, receipt of mechanical ventilation, and recent cardiac arrest [22], was used. A Pitt bacteremia score (PBS) of 0 was regarded as stabilized illness, whereas 1 to 3 and ≥4 were regarded as moderate and critical illness, respectively [19,22].

Malignancies included hematological malignancies and solid tumors. Comorbidities were defined as described previously [23], and the comorbid severity was assessed by the McCabe classification [24]. The sources of bacteremia were determined clinically based on the presence of an active infection site coincident with bacteremia or the isolation of a microorganism from other clinical specimens prior to, or on the same date of, bacteremia onset. If the source of bacteremia could not be traced to a specific site, it was classified as primary bacteremia. The occurrence of an EKP bacteremic episode, despite the administration of in vitro active agents for at least 24 h, was regarded as breakthrough bacteremia [25]; the re-emergence of bloodstream infection due to the same pathogen in follow-up blood cultures after the discontinuation of appropriate antimicrobials was referred to be recurrent bacteremia [26]. Consistent with a previous definition, the removal of infected hardware, drainage of infected fluid collection, or resolution of the obstruction of biliary or urinary sources, was considered to be appropriate source control [27]. Crude mortality was used to define death from all causes. 

### 2.6. Statistical Analysis

Statistical analyses were performed by the Statistical Package for the Social Science for Windows (Chicago, IL, USA), version 23.0. Continuous variables were expressed as the mean values ± standard deviations or medians (interquartile ranges, IQRs) and compared by the Student’s *t* test. Categorical variables were expressed as numbers and percentages and compared by the Chi-square or Fisher’s exact test. A linear-by-linear association of cefazolin MICs and clinical variables was studied by the *Spearman’s* correlation, presented by *Spearman’s* rho (*γ*, correlation coefficients) and *p* values.

To recognize the independent predictors, all predictors of 30-day mortality with a *p* value less than 0.1 in the univariate analysis were included in a stepwise and backward multivariable logistic regression model. Kaplan–Meier survival curves, analyzed by the Cox proportional hazard model after adjustment for independent predictors were used to compare the effects of varied cefazolin MICs on treatment failure or 30-day mortality. A two-tailed *p* value of less than 0.05 was considered significant.

## 3. Results

### 3.1. Demographics and Clinical Characteristics

A total of 446 adults with community-onset monomicrobial EKP bacteremia and definitively treated by intravenous cefazolin were included based on the inclusion and exclusion criteria (Figure 1). Their mean age was 67.8 years and 305 (68.4%) were female. The proportion of critically ill patients (PBS ≥ 4) at onset and Day 3 of bacteremic episodes was 9.2% (41 patients) and 6.1% (27), respectively; the stabilized patients (PBS = 0) at bacteremia onset and Day 3 accounted for 31.2% (139 patients) and 78.3% (349), respectively. The median (IQR) duration of intravenous cefazolin therapy and hospitalization was 7 (6–11) days and 9 (7–13) days, respectively. A crude 15- and 30-day mortality rate was 0.7% (three patients) and 2.2% (10), respectively. Fifty patients experiencing treatment failure of definitive cefazolin therapy accounted for 11.2% of the entire cohort.

Common sources of bacteremia included urinary tract infections (293 patients, 65.7%), biliary tract infections (40, 9.0%), intra-abdominal infections (36, 8.1%), liver abscess (24, 5.4%), pneumonia (12, 2.7%), skin and soft-tissue infections (five, 1.1%), as well as bone and joint infections (five, 1.1%). Primary bacteremia accounted for only 7.0% (31 patients). Of 446 causative microorganisms, there were 340 (76.2%) *E. coli*, 90 (20.2%) *Klebsiella* species (including 89 *Klebsiella pneumoniae* and one *Klebsiella oxytoca*), and 16 (3.6%) *P. mirabilis* isolates. The leading rate of 15- and 30-day crude mortality was 6.3% (one patient) and 12.5% (two) in patients infected by *P. mirabilis*, followed those by *Klebsiella* species (1.1% (one patient) and 4.4% (four)) and *E. coli* (0.3% (one) and 1.2% (four)), respectively.

### 3.2. Clinical Characteristics and Outcomes in Varied Cefazolin MIC Groups

The trends in clinical characteristics, in terms of demographics, type and severity of comorbidities, bacteremia sources, bacteremia severity, types of empirical antimicrobials, and patient outcomes, in different cefazolin MIC groups, were shown in Table 1. A negative, MIC-related trend was observed only in the proportions of comorbid neurological diseases and stabilized patients (PBS = 0) at bacteremia onset. Furthermore, positive MIC-related trends in the duration of intravenous and total antimicrobial administration could be disclosed. In terms of patient outcomes, as cefazolin MICs increased, the case numbers of treatment failure, the 30-day crude mortality rate after bacteremia onset, the proportions of recurrent bacteremia, and the 90-day crude mortality rate after the end of definitive cefazolin therapy increased (Figure 2; all *γ* = 1.00, *p* = 0.01).

### 3.3. Risk Factors of Treatment Failure of Definitive Cefazolin Therapy

Clinical variables negatively or positively associated with treatment failure under definitive cefazolin therapy, such as stabilized status (PBS = 0) at Day 3, underlying diabetes mellitus, and *E. coli* bacteremia, or rapidly or ultimately fatal comorbidity (McCabe classification) and malignancies, were evident in the univariate analysis (Table 2). In the multivariate regression, a protective variable (stabilized status at Day 3) and a predictive variable (rapidly or ultimately fatal comorbidity) for treatment failure were identified.

### 3.4. Impact of Cefazolin MICs on Treatment Failure and 30-Day Crude Mortality

Of 446 eligible patients, the therapeutic efficacy, as evidenced by the treatment failure of definitive cefazolin therapy, categorized by different breakpoints of cefazolin MICs, i.e., 1, 2, 4, or 8 mg/L, was shown in the Kaplan–Meier curves using the Cox proportional hazard model after adjustment for two independent determinants of treatment failure (Figure 3A). A significant impact was evidenced in the breakpoints of 2 mg/L (adjusted hazard ratio [AHR], 1.93; *p* < 0.001) and 4 mg/L (AHR, 2.02; *p* < 0.001).

Two independent variables of 30-day crude mortality were identified by the multivariate regression analysis (Table 3). One protective factor was the presence of stabilized status (PBS = 0) at Day 3, and a positive predictor was rapidly or ultimately fatal comorbidity. In the Kaplan–Meier curves analyzed by the Cox proportional hazard model, with adjustment for two independent determinants, the impact of cefazolin MICs on 30-day crude morality was most evident in the MIC breakpoint of 2 mg/L (AHR, 4.79; *p* = 0.02; Figure 3B).

## 4. Discussion

The cefazolin interpretive criteria for Enterobacteriaceae of CLSI have been revised since 2010. In 2010 and 2012, CLSI released recommended susceptible breakpoints for cefazolin, namely MIC ≤ 1 mg/L and ≤2 mg/L, respectively. This modification was reported to be largely based on relevant in vitro susceptibility data and pharmacokinetic–pharmacodynamic information [11,12]. In our cohort, the significant impact on 30-day mortality was observed in patients with bacteremia caused by EKP isolates with cefazolin MIC of >2 mg/L, compared to those of ≤2 mg/L. Therefore, we provided clinical evidence supporting inadequate therapeutic efficacies of definitive cefazolin therapy for bloodstream infections caused by “cefazolin-intermediate” (i.e., MIC = 4 mg/L) or “cefazolin-resistant” (MIC ≥ 8 mg/L) EKP isolates, according to the contemporary CLSI criteria. Consistent with a recent review [28], the association of antimicrobial MICs and clinical outcomes in patients with Gram-negative bacilli bacteremia was highlighted. Moreover, a linear-by-linear association of therapeutic efficacy (i.e., treatment failure) and cefazolin MICs was demonstrated.

As pointed out by Tamma et al. in 2014 [29], not all microbiology laboratories in the United States followed the updated CLSI cephalosporin breakpoints. Our microbiology laboratory adopted revised cefazolin breakpoints for Enterobacteriaceae isolates in August 2015. Thus, we could include 52 patients with bacteremia caused by cefazolin-nonsusceptible (MIC = 4–16 mg/L) EKP isolates who had received definitive cefazolin treatment.

Clinical outcomes in our cohort should be cautiously interpreted. In general, those who did not die within three days after bacteremia onset and only hospitalized patients were included. Only those with a less critical illness at bacteremia onset were eligible here. Accordingly, the efficacies of definitive cefazolin therapy presented here may not be generalized to other populations. Furthermore, the majority (78%, 349/446) of patients with definitive cefazolin therapy were stabilized after 72 h of appropriate empirical therapy. Accordingly, focusing on the aimed patients definitively treated by cefazolin in our cohort, it should be considered that the statistical power might be substantially limited because of the low short-term mortality rate. Therefore, another composite parameter (i.e., treatment failure) to assess the efficacies of definitive cefazolin therapy was included in our analyses. With the outcome information, such as treatment failure or 30-day mortality rate, the reasonability of the contemporary MIC breakpoint for cefazolin susceptibility was rightfully highlighted here.

Currently, the therapeutic role of cefazolin has been emphasized on surgical prophylaxis [5] and the treatment of infectious diseases due to cefazolin-susceptible, gram-positive organisms [7,8]. Although it was active against common pathogens in the community, such as *E. coli*, *K. pneumoniae*, and *P. mirabilis*, clinical outcome information dealing with these common bacterial infections, except for urinary tract infections [30,31,32], was limited. Recently, numerous investigations focused on those with systemic infections, especially bacteremia or septicemia [14,15,33,34], but the rationale of the revised MIC breakpoint to augment cefazolin administration as definitive therapy remained under debate. Furthermore, antimicrobial resistance in bacterial microorganism is a worldwide challenge, resulting in high morbidity and mortality [35]. To minimize the use of broad-spectrum antibiotics in the era of increasing antimicrobial resistance among the pathogens causing community-acquired or healthcare-associated infections, it is crucial to explore the therapeutic role of intravenous cefazolin for adults with EKP bacteremia. Although patients empirically treated by various active antimicrobials were included, our study showed that definitive cefazolin therapy was safe for those infected by EKP isolates with cefazolin MICs of ≤2 mg/L, if a patient is stable after 72 hours’ empirical antimicrobial therapy.

As the disease severity of bloodstream infections adversely affects the clinical outcome of affected patients, clinical grading of sepsis severity is essential in accessing the therapeutic efficacy of different antimicrobial regimens. There were two reasons for choosing the simple Pitt bacteremia score as the indicator of bacteremia severity. Firstly, the clinical validity of the PBS was well demonstrated in community-onset bacteremia [4,19] and specific microorganisms, particularly in Enterobacteriaceae [22,36]. Secondly, a low PBS (= 0) indicative of a stable clinical condition was reported in the previous studies discussing the patient cohort with community-onset bloodstream infections [19].

There are some limitations inherent in the design of this study. Firstly, it is a retrospective study conducted in two hospitals. However, it is unethical to prospectively assign patients to be treated by cefazolin to assess their clinical outcomes, especially for the treatment of cefazolin-resistant or -intermediate EKP based on updated breakpoints. To avoid ethical conflicts, it was reasonable to retrospectively review medical records and patient outcomes. Secondly, to assess the therapeutic efficacy of antimicrobial therapy, those with incomplete outcome information or who had received inappropriate empirical antimicrobial therapy were excluded. Although this could result in an underestimation of the mortality rate, due to the lack of access to critically or fatally ill patients, only a small proportion of the entire cohort was excluded. Such a bias in patient selection would be trivial. Finally, the diversity of empirical antimicrobial agents in the varied cefazolin groups was not considered, but the appropriateness of empirical therapy was controlled for in our population.

## 5. Conclusions

For adults with community-onset bacteremia due to an EKP isolate with cefazolin MIC ≤2 mg/L, definitive cefazolin therapy can result in a favorable prognosis, which is consistent with the contemporary susceptible breakpoint of CLSI. Accordingly, antimicrobial de-escalation to cefazolin can be considered in the treatment of bacteremia caused by “cefazolin-susceptible” EKP isolates in stabilized adults.

## Figures and Tables

**Figure 1 jcm-09-00157-f001:**
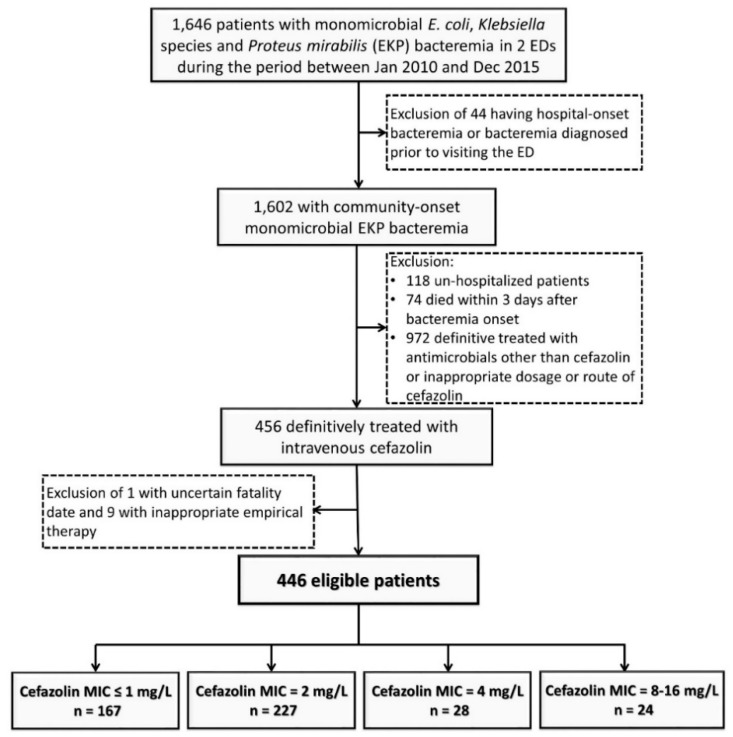
Flowchart of patient selection. ED = emergency department; MIC = minimum inhibitory concentration.

**Figure 2 jcm-09-00157-f002:**
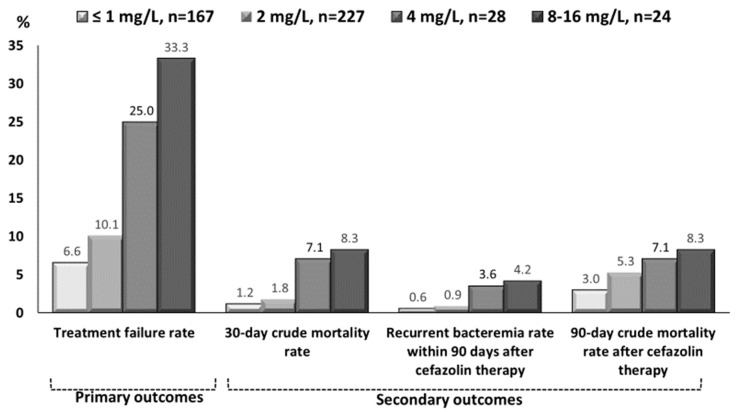
The cefazolin-MIC-related trend (all *γ* = 1.00, *p* = 0.01) in primary and secondary outcomes of adults with community-onset monomicrobial *Escherichia coli*, *Klebsiella* species, or *Proteus mirabilis* bacteremia definitively treated by cefazolin. Early treatment failure, i.e., primary outcome, was the composite of antimicrobial escalation to broad-spectrum agents, the development of breakthrough bacteremia, the need for intensive care during definitive cefazolin therapy, and crude mortality within 15 days after bacteremia onset.

**Figure 3 jcm-09-00157-f003:**
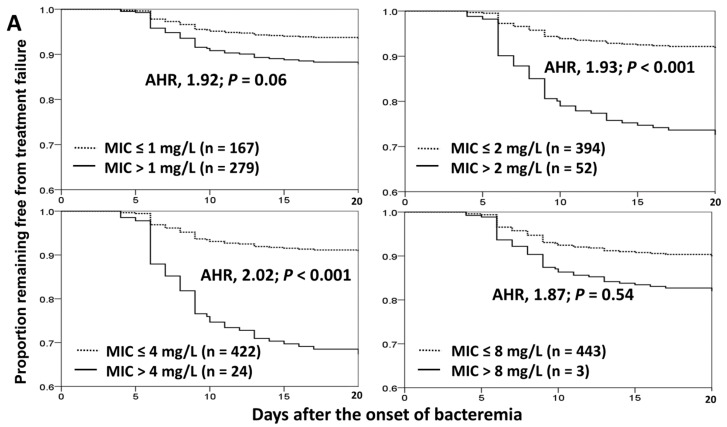
Kaplan–Meier curves in treatment failure of definitive cefazolin therapy (**A**) or 30-day crude mortality after bacteremia onset (**B**), categorized by different interpretative breakpoints of cefazolin MICs using the Cox proportional hazard model. AHR = adjusted hazard ratio; MIC = minimum inhibitory concentration. The adjusting independent predictors included a stabilized illness (a Pitt bacteremia score = 0) at Day 3 and an ultimately or rapidly fatal comorbidity (McCabe classification) respectively in (**A**,**B**).

**Table 1 jcm-09-00157-t001:** Clinical characteristics and outcomes of 446 adults with community-onset monomicrobial *Escherichia coli*, *Klebsiella* species, or *Proteus mirabilis* bacteremia definitively treated by cefazolin, categorized by cefazolin MICs.

Characteristics	Patient Number (%)				*γ*	*p* Values
	≤1 mg/L, *n* = 167	2 mg/L, *n* = 227	4 mg/L, *n* = 28	8–16 mg/L, *n* = 24		
Gender, female	116 (69.5)	158 (69.6)	18 (64.3)	13 (54.2)	−0.80	0.20
Old age, ≥65 years	110 (65.9)	131 (57.7)	21 (75.0)	18 (75.0)	0.74	0.26
Nursing-home residents	1 (0.6)	2 (0.9)	0 (0)	0 (0)	−0.74	0.26
Major comorbidities						
Hypertension	92 (55.1)	110 (48.5)	19 (67.9)	16 (66.7)	0.60	0.40
Diabetes mellitus	72 (43.1)	101 (44.5)	13 (46.4)	6 (25.0)	−0.20	0.80
Malignancies	37 (22.2)	48 (21.1)	9 (32.1)	5 (20.8)	−0.40	0.60
**Neurological diseases**	**35 (21.0)**	**44 (19.4)**	**4 (14.3)**	**2 (8.3)**	**−1.00**	**0.01**
Chronic kidney diseases	21 (12.6)	21 (9.3)	6 (21.4)	6 (25.0)	0.80	0.20
Liver cirrhosis	19 (11.4)	31 (13.7)	2 (7.1)	4 (16.7)	0.40	0.60
Major source of bacteremia						
Urinary tract	109 (65.3)	147 (64.8)	21 (75.0)	16 (66.7)	0.60	0.40
Biliary tract	20 (12.0)	12 (5.3)	6 (21.4)	2 (8.3)	0	1.00
Intra-abdominal	19 (11.4)	16 (7.0)	0 (0)	1 (4.2)	−0.80	0.20
Primary bacteremia	13 (7.8)	14 (6.2)	1 (3.6)	3 (12.5)	0.20	0.80
Pneumonia	3 (1.8)	7 (3.1)	0 (0)	2 (8.3)	0.40	0.60
Liver abscess	1 (0.6)	23 (10.1)	0 (0)	0 (0)	−0.74	0.26
Rapidly or ultimately fatal comorbidities (McCabe classification)	24 (14.4)	36 (15.9)	4 (14.3)	11 (45.8)	0.40	0.60
Inadequate source control during antibiotic therapy	3 (1.8)	9 (4.0)	0 (0)	1 (4.2)	0.40	0.60
Pitt bacteremia score						
Onset						
**0**	**57 (34.1)**	**70 (30.8)**	**7 (25.0)**	**5 (20.8)**	**−1.00**	**0.01**
≥4	19 (11.4)	21 (9.3)	0 (0)	1 (4.2)	−0.80	0.20
Day 3 after onset						
0	139(83.2)	168 (74.0)	23 (82.1)	19 (79.2)	−0.40	0.60
≥4	13 (7.8)	13 (5.7)	0 (0)	1 (4.2)	−0.80	0.20
Type of empirical antibiotics						
Third-generation cephalosporins	85 (50.9)	121 (53.3)	11 (39.3)	13 (54.2)	0.40	0.60
First-generation cephalosporins	31 (18.6)	45 (19.8)	3 (10.7)	3 (12.5)	−0.60	0.40
Second-generation cephalosporins	26 (15.6)	34 (15.0)	7 (25.0)	4 (16.7)	0.60	0.40
Fluoroquinolones	9 (5.4)	7 (3.1)	1 (3.6)	2 (8.3)	0.40	0.60
Ampicillin/sulbactam	8 (4.8)	3 (1.3)	3 (10.7)	0 (0)	−0.40	0.60
Fourth-generation cephalosporins	4 (2.4)	10 (4.4)	2 (7.1)	1 (4.2)	0.40	0.60
Carbapenems	4 (2.4)	4 (1.8)	1 (3.6)	0 (0)	−0.40	0.60
Others	0 (0)	3 (1.3)	0 (0)	1 (4.2)	0.63	0.37
Duration (mean ± standard deviation)						
Time-to-appropriate antibiotic, hour *	3.2 ± 5.6	2.4 ± 3.0	4.8 ± 8.1	4.5 ± 5.6	0.60	0.40
Time-to-cefazolin therapy, day **	2.8 ± 1.9	2.7 ± 2.1	3.3 ± 2.2	3.0 ± 1.4	0.60	0.40
Intravenous cefazolin therapy, day	5.0 ± 3.5	5.4 ± 3.9	6.0 ± 3.6	5.7 ± 3.4	0.80	0.20
**Intravenous antimicrobial therapy, day**	**8.7 ± 5.4**	**9.0 ± 6.5**	**11.7± 7.9**	**12.0 ± 5.3**	**1.00**	**0.01**
**Total antibiotic administration, day**	**12.4 ± 4.8**	**13.1 ± 5.5**	**15.9 ± 6.9**	**16.7 ± 5.2**	**1.00**	**0.01**
**Length of hospitalization among the survivors, day**	**10.7 ± 6.6**	**11.3 ± 10.2**	**13.2 ± 8.1**	**14.1 ± 5.3**	**1.00**	**0.01**
Primary outcomes (i.e., Treatment failure)						
**Overall**	**11 (6.6)**	**23 (10.1)**	**7 (25.0)**	**8 (33.3)**	**1.00**	**0.01**
**Escalation to broad-spectrum during cefazolin therapy**	**11 (6.6)**	**20 (8.8)**	**7 (25.0)**	**7 (29.2)**	**1.00**	**0.01**
Breakthrough bacteremia during cefazolin therapy	0 (0)	1 (0.4)	1 (3.6)	0 (0)	0.11	0.90
Transfer to intensive care unit during cefazolin therapy	0 (0)	2 (0.8)	0 (0)	0 (0)	−0.26	0.74
**15-day crude mortality after bacteremia onset**	**0 (0)**	**1 (0.4)**	**1 (3.6)**	**1 (4.2)**	**1.00**	**0.01**
Secondary outcomes						
**30-day crude mortality rate after bacteremia onset**	**2 (1.2)**	**4 (1.8)**	**2 (7.1)**	**2 (8.3)**	**1.00**	**0.01**
**Recurrent bacteremia within 90 days after cefazolin therapy**	**1 (0.6)**	**2 (0.9)**	**1 (3.6)**	**1 (4.2)**	**1.00**	**0.01**
**90-day crude mortality after cefazolin therapy**	**5 (3.0)**	**12 (5.3)**	**2 (7.1)**	**2 (8.3)**	**1.00**	**0.01**

Data are given as number (percent), unless otherwise specified. Boldface indicates statistical significance, i.e., a *p* value of <0.05. *γ* indicates *Spearman* correlation coefficients. * The period between bacteremia onset and administration of appropriate antimicrobials. ** The period between bacteremia onset and cefazolin administration.

**Table 2 jcm-09-00157-t002:** Risk factors of treatment failure of definitive cefazolin therapy.

Variable	Patient Number (%)		Univariate Analysis		Multivariate Analysis	
	Failure, *n* = 49	Success, *n* = 396	OR (95% CI)	*p* Value	Adjusted OR (95%CI)	*p* Value
Pitt bacteremia score = 0 at Day 3	31 (63.3)	318 (80.1)	0.43 (0.23–0.80)	0.007	0.43 (0.23–0.81)	0.009
Rapidly or ultimately fatal comorbidities (McCabe classification)	14 (28.6)	61 (15.4)	2.20 (1.12–4.34)	0.02	2.21 (1.12–4.39)	0.02
Comorbid malignancies	17 (34.7)	82 (20.7)	2.04 (1.08–3.86)	0.04	NS	NS
Bacteremia due to urinary tract infections	27 (55.1)	266 (67.0)	0.60 (0.33–1.10)	0.098	NS	NS
Causative microorganisms						
*Escherichia coli*	29 (59.2)	311 (78.3)	0.40 (0.22–0.74)	0.003	NS	NS
*Proteus mirabilis*	5 (10.2)	11 (2.8)	3.99 (1.32–12.01)	0.02	NS	NS

NS = No significance (after processing the stepwise and backward multivariate regression); OR = odds ratio; CI = confidence interval.

**Table 3 jcm-09-00157-t003:** Risk factors of 30-day crude mortality.

Variable	Patient Number (%)		Univariate Analysis		Multivariate Analysis	
	Death, *n* = 10	Survival, *n* = 436	OR (95% CI)	*p* Value	Adjusted OR (95% CI)	*p* Value
Pitt bacteremia score						
0 at onset	0 (0)	139 (31.9)	-	0.04	NS	NS
0 at Day 3	1 (10.0)	348 (79.8)	0.03 (0.004–0.23)	<0.001	0.03 (0.004–0.25)	0.001
Sources of bacteremia						
Pneumonia	2 (20.0)	10 (2.3)	10.65 (2.00–56.66)	0.001	NS	NS
Urinary tract infections	4 (40.0)	289 (66.3)	0.34 (0.09–1.22)	0.099	NS	NS
Causative microorganisms						
*Escherichia coli*	4 (40.0)	336 (77.1)	0.20 (0.06–0.72)	0.01	NS	NS
*Proteus mirabilis*	2 (20.0)	14 (3.2)	7.54 (1.46–38.79)	0.046	NS	NS
Rapidly or ultimately fatal comorbidities (McCabe classification)	6 (60.0)	69 (15.8)	7.98 (2.19–29.01)	0.002	7.55 (1.82–31.31)	0.005
Comorbidity types						
Diabetes mellitus	0 (0)	192 (44.0)	-	0.006	NS	NS
Malignancies	7 (70.0)	92 (21.1)	8.73 (2.21–34.4)	0.001	NS	NS

NS = No significance (after processing the stepwise and backward multivariate regression); OR = odds ratio; CI = confidence interval.

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
