# Peer review of "Definitive Cefazolin Therapy for Stabilized Adults with Community-Onset *Escherichia coli*, *Klebsiella* Species, and *Proteus mirabilis* Bacteremia: MIC Matters"

_jcm, 2020, doi:10.3390/jcm9010157_

Round 1

Reviewer 1 Report

JCM-668163

Hsieh et al. Definitive cefazolin therapy for stabilized adults with community-onset Escherichia coli, Klebsiella species, and Proteus mirabilis bactermia: MIC matters.

This paper presents important data regarding the results of treatment with cefazolin in Enterobacteriacae bacteremia, which will be a valuable resource for use in determining cefazolin breakpoints.

Once the results are clear on the following three issues, I think the arguments in this paper will be better substantiated and more clearly interpretable.

It is necessary to explain why EKP was chosen for the study in the Introduction or Discussion.

Wouldn’t it be better to employ regression analysis instead of correlation analysis to observe the causal relationship between MIC, treatment outcomes, and mortality?

The authors have published another paper using the same cohort group (Antibiotics 2019,8(4),216). In that paper, the cefazolin group had few underlying conditions, with low average severity, and cefazolin was mainly used meaningfully for urinary tract infections. Demonstration of susceptibility alone, therefore, is insufficient to conclude that cefazolin can be useful in EKP bacteremia. Moreover, in this study, the significant risk factors for treatment outcomes and 30-day mortality rates were the severity of bacteremia during hospital visits (Pitt bacteremia score) and the patient’s underlying disease (McCabe classification). To determine whether the MIC 2 criterion can be judged meaningfully, it would be better to divide the two groups based on MIC 2 and calibrate other variables through regression analysis.

Author Response

Hsieh et al. Definitive cefazolin therapy for stabilized adults with community-onset Escherichia coli, Klebsiella species, and Proteus mirabilis bacteremia: MIC matters.This paper presents important data regarding the results of treatment with cefazolin in Enterobacteriaceae bacteremia, which will be a valuable resource for use in determining cefazolin breakpoints.Response: Many thanks for your review.

Once the results are clear on the following three issues, I think the arguments in this paper will be better substantiated and more clearly interpretable.

 (1) It is necessary to explain why EKP was chosen for the study in the Introduction or Discussion.

Response: Thanks for your suggestion. The related sentences and references were revised in the section of "Introduction" (Line 46-48, Page 2).

(2) Wouldn’t it be better to employ regression analysis instead of correlation analysis to observe the causal relationship between MIC, treatment outcomes, and mortality?

Response: Thanks for your question. In our study design, to demonstrate the MIC-related trends in the real world (such as bacteremia severity, comorbidity severity, and the duration of antimicrobial administration), a linear-by-linear association of cefazolin MICs and indicated variables was presented by the Spearman’s correlation.

Furthermore, to study the impact of varied cefazolin MICs on treatment failure (Figure 3A) or 30-day mortality (Figure 3B), the Kaplan-Meier survival curves combined with the Cox proportional hazard model was more suitable than the performance of the logistic regression analysis for the following two reasons

Similar to the logistic regression model, the Cox proportional hazard model is essentially a regression model commonly used statistical in medical research for investigating the association between the survival time of patients and one or more predictor variables, after adjustment of confounding factors. In our cohort, the all independent predictors of treatment failure and 30-day mortality respectively recognized by the logistic regression model, as shown in Table 2 and 3, were adjusted using the Cox proportional hazard model

Superior to the logistic regression model, the proportion of fatal cases that develop in a given time period was considered in Kaplan-Meier curves combined with the Cox proportional hazard model; in other words, the factor of "time to death" was only assessed in the assigned model herein.

(3) The authors have published another paper using the same cohort group (Antibiotics 2019,8(4),216). In that paper, the cefazolin group had few underlying conditions, with low average severity, and cefazolin was mainly used meaningfully for urinary tract infections. Demonstration of susceptibility alone, therefore, is insufficient to conclude that cefazolin can be useful in EKP bacteremia. Moreover, in this study, the significant risk factors for treatment outcomes and 30-day mortality rates were the severity of bacteremia during hospital visits (Pitt bacteremia score) and the patient’s underlying disease (McCabe classification). To determine whether the MIC 2 criterion can be judged meaningfully, it would be better to divide the two groups based on MIC 2 and calibrate other variables through regression analysis.

Response: Thanks for your opinions and suggestions. This 2-center, 6-year study cohort differed from the previously published cohort (Antibiotics 2019,8(4),216) involving one medical center during the 8-year period. Additionally, the aimed population was also different between the present cohort (i.e., patients definitively treated with cefazolin) and the previously published cohort (i.e., patients infected by cefazolin-susceptible EKP); the goal of study was respectively to demonstrate the impact of revised CLSI breakpoints on patient outcomes in the current cohort and to compare the efficacy of definitive cefazolin with other broader-spectrum antimicrobials in the published cohort.

As above statement, the Kaplan-Meier curves combined with the Cox proportional hazard model was more appropriate to investigate the impact of varied MIC breakpoints (including MIC ≤ 2 vs. MIC >2) on patient outcomes after adjustment of all independent predictors (including bacteremia severity [a Pitt bacteremia score] and comorbidity severity [the McCabe classification]) recognized by the logistic regression model, shown in Figure 3A and 3B.

Reviewer 2 Report

There are some resent papers about the asian and south asian situation that should be referenced in the introduction. And row 57-59 is not true but a misunderstanding from the Authors and need to be corrected by contacting Eucast. The Eucast group have performed this but due to different reasons not published, in a easy way.

The method section is very well written and shows what have been done. with good references.

IN THE RESULT SECTION A BETTER DESCRIPTION WHAT IS INCLUDED IN CRUDE MORTALLITY IS NEEDED. This since it do not completely seem to follow standard way of what crudemortallity is.

A general comment is that it would have been interesting to know what sub species bacterias such of klebsiella etc is studied. Othervise the result section is good for publication even if it can be even more clear.

The first well known references are good but old the field of AMR is changing rapidly and therefor an update with some more recent references is needed in my opinion, this to show both the readers and the world that the authors have a complete understanding of the situation and the emergent out break in countries close to Taiwan such as china. 

Author Response

There are some recent papers about the Asian and south Asian situation that should be referenced in the introduction. And row 57-59 is not true but a misunderstanding from the Authors and need to be corrected by contacting EUCAST. The EUCAST group have performed this but due to different reasons not published, in an easy way.Response: Thanks for your opinions. Two new references (Ref No. 14 and 15) dealing with issue about the alteration of interpretive breakpoints and the related sentences (Line 62-64, Page 2) were inserted. Moreover, the sentence about the EUCAST statement was also revised (Line 57-58, Page 2).

The method section is very well written and shows what have been done. with good references.Response: Many thanks for your review.

IN THE RESULT SECTION A BETTER DESCRIPTION WHAT IS INCLUDED IN CRUDE MORTALITY IS NEEDED. This since it does not completely seem to follow standard way of what crude mortality is.A general comment is that it would have been interesting to know what sub species bacteria such of klebsiella etc. is studied. Otherwise the result section is good for publication even if it can be even more clear.Response: Thanks for your suggestions. The definition of crude mortality was descried in Line 141-142 on Page 4. The crude mortality rate in the microorganism subgroup was in detail inserted in the section of "Result" (Line 176-178, Page 5). Additionally, the species distribution of klebsiella genus was revised in Line 175-176 (Page 5). Because K. pneumoniae accounted for extreme majority (89/90, 98.9%) of klebsiella genus, it was difficult to study the differences between two species.

The first well known references are good but old the field of AMR is changing rapidly and therefore an update with some more recent references is needed in my opinion, this to show both the readers and the world that the authors have a complete understanding of the situation and the emergent outbreak in countries close to Taiwan such as China. Response: Thanks for your suggestions. Several older references were deleted (Line 239-240, Page 9) and numerous updated references (Ref. No. 14, 15, and 34) discussed with the efficacies of cefazolin therapy for EKP bacteremia were inserted in the revised manuscript. Additionally, a new reference (Ref. No. 35) emphasized the increasing AMR worldwide was also added. These revised sentences were listed in Line 269-273 (Page 10).